# Diamagnetically levitated nanopositioners with large-range and multiple degrees of freedom

K. S. Vikrant [1] & G. R. Jayanth [1,2✉]

Precision positioning stages are often central to science and technology at the micrometer and nanometer length scales. Compact, multi-degree-of-freedom stages with large dynamic range are especially desirable, since they help to improve the throughput and versatility in manipulation without introducing spatial constraints. Here, we report compact diamagnetically levitated stages, which employ dual-sided actuation to achieve large-range, six degrees-of-freedom positioning. Dual-sided actuation is demonstrated to enable trapping a magnet array in 3D, with independent control of the trap stiffness about two axes, independent control of forces in 3D and torque about 2 axes. A simplified model is proposed to directly relate these physical quantities to the necessary actuation currents. Experimentally, we demonstrate six degrees-of-freedom positioning with low cross-axis motion, large range and nanometer-scale resolution. In particular, here we show linear motion range of 5 mm with positioning precision better than 1.88 nm, and angular motion range of 1.1 radian with a resolution of 50 micro-radian. With the volume of the stage being between 10-20 cm$^3$, its utility as a compact nano-positioner is showcased by using it to automatically replace the tip of an atomic force microscope probe.

[1] Department of Instrumentation and Applied Physics, Indian Institute of Science, Bangalore 560012, India. [2] Department of Mechanical Engineering, Indian Institute of Science, Bangalore 560012, India. ✉email: jayanth@iisc.ac.in

Precision positioning stages are indispensable in many areas of science and engineering involving manipulation[1–4], fabrication[5–9], imaging[10–13], material characterization[14–17], and force control[18–21], with applications in biology, medicine, manufacturing, robotics, and microscopy. The specific technology adopted for positioning depends on the necessary range and resolution requirements, and generally necessitates making compromises in one of them. Since magnetically levitated stages do not experience dry friction, they are among the few technologies that can simultaneously achieve high range, on the order of millimeters along multiple degrees of freedom and high resolution, on the order of nanometers[22–25]. Conventionally, the stages are levitated actively by employing precision measurement, actuation, and control systems. Active magnetic levitation has found applications in micro-fabrication[26,27], manipulation[28–30], lithography[22,31], metrology[32–35], and microscopy[36,37]. However, the measurement and actuation subsystems contribute to a relatively large footprint for the stage, making it suitable primarily for use as a standalone instrument. Furthermore, while linear motion ranges are often large, angular ranges are small on account of the stage design. An alternative to active levitation is passive levitation, where an array of permanent magnets is levitated over a diamagnetic material, such as graphite, and is actuated using current-carrying traces situated on one side of the magnet array and typically patterned on a printed circuit board (PCB)[38]. The magnet array is first trapped at a specific lateral position above the traces by the applied currents and are displaced by appropriately changing the currents in the traces. Since diamagnetic levitation is intrinsically stable[39–41], the necessary measurement and control subsystems can be significantly simpler, leading to more compact systems, with small thickness and much lesser volume than those of active stages.

Diamagnetically levitated magnet arrays have been previously employed as robotic platforms to undertake a variety of manipulation tasks[42–45]. The magnet arrays can be moved over centimeter-scale distances in the plane of the traces, with a repeatability of about 200 nm[38], and by dividing the traces into zones, a single array has been rotated in-plane[46] by about ±10°. However, the reported repeatability is not adequate for nanopositioning applications. Furthermore, the use of single-sided current-carrying traces results in unavoidable coupling in actuation, between the in-plane forces and the in-plane moments, as also between the in-plane stiffness and out-of-plane magnetic force. The latter coupling, in particular, leads to reduced levitation heights, with consequent limits on payload capabilities, and on the achievable stiffnesses with which the magnets can be trapped. The relatively low stiffness, in turn, leads to increased sensitivity to vibration.

In this work, we propose dual-sided actuation, wherein actuating traces are located symmetrically both above and below the levitated magnet array. We show that the proposed arrangement enables independent control of seven quantities, namely, forces in X-, Y- and Z-directions, torques about X- and Y-axes, and linear stiffnesses along X- and Y- axes. It is also shown that the loads experienced by the array is nearly equivalent to that of a point-dipole magnet. This has been used to obtain simple relations between the actuating currents and the desired forces, moments and stiffnesses. We also demonstrate that forces and torque are primarily limited by the achievable actuation currents while the achievable stiffnesses along all three axes are ultimately limited by the stiffness of diamagnetic interaction. Next, we show that by using four separate zones of actuating traces, it is possible to actuate the magnet array along all six degrees-of-freedom and achieve large range for rotations about the Z-axis. Likewise, by integrating them with a compliant trapezoidal mechanism, we develop a six degrees-of-freedom

stage with large linear motion range along the Z-axis. Due to its small form factor and excellent positioning capability, the developed positioners can be retrofitted into an Atomic Force Microscope (AFM). Here, its use for automatically replacing tips of the AFM is demonstrated.

Dual-sided actuation demonstrates between one and two orders of magnitude improvements in performance over single-sided actuation: we report positioning stability better than 2 nm root-mean-square (RMS), i.e., over 100-fold improvement, cross-axis motion of about 2 μm peak-to-peak, i.e., over 5-fold reduction, out-of-plane positioning range of about 900 μm, i.e., about 18-fold increase in range and arbitrarily large in-plane angular positioning range, though here we have demonstrated a range of ±31.5°, i.e., over 3.1-fold increase in in-plane angular positioning range, and with a positioning resolution of 50 μrad. In short, the proposed designs achieve comparable positioning performance as actively levitated stages along the linear displacement channels even without feedback control, but with the actuator volume being at least 10 times lesser. For in-plane rotations, unlike active stages which typically achieve milliradian-scale range, here, arbitrarily large ranges are possible but with similar positioning resolutions.

The rest of the paper is divided as follows: the results section describes the principle, theoretical analysis, and experimental results with both the six degrees-of-freedom stages. Finally, the application of the positioning stage for automated tip replacement in AFM is presented. The methods section describes hardware and software details of the set-up.

## Results

**Design of the diamagnetically levitated magnetic actuator**. A schematic of the proposed diamagnetically levitated actuator is shown in Fig. 1a. The system comprises a magnet array sandwiched between two identical actuator traces, both patterned on printed circuit boards (PCBs), and with a pyrolytic graphite plate positioned above the bottom traces. The magnet array is a chequerboard of identical square-shaped permanent magnets with alternating upward and downward magnetic moments positioned from each other with a pitch $p$ along the X- and Y-axes. The pyrolytic graphite plate is of thickness $t_d$ and is used to diamagnetically levitate the magnet array in a plane at a height $z_d$ above it (see Supplementary Note 1 for obtaining $z_d$). Each actuating side carries two pairs of traces, with one pair aligned along the Y-axis and the other along the X-axis, and the offset between two adjacent traces of a pair being $p/4$. Each trace comprises straight conductors arranged in a meandering fashion with the pitch of the meander being $p$. The gap $z_0$ between the levitating plane and the traces is chosen to be $z_0 = z_d + t_d$, which ensures that the center of the levitating magnet array is positioned symmetrically between the two actuating sides. Furthermore, the pitch is chosen such that the height $z_0$ of the magnets above the bottom actuator is comparable to the pitch $p$ of the wires. The traces on the two sides together provide eight independent currents to control the magnet array, namely, the X-currents of the upper PCB $I_x^{u1}$, $I_x^{u2}$, the X-currents of the lower PCB $I_x^{l1}$, $I_x^{l2}$, the Y-currents of the upper PCB $I_y^{u1}$, $I_y^{u2}$ and the Y-currents of the lower PCB $I_y^{l1}$, $I_y^{l2}$. These eight currents, between them, enable independent control of seven quantities, namely, the forces $\mathbf{F} = [F_x \; F_y \; F_z]^T$ along X-, Y- and Z- axes, the moments $\boldsymbol{\tau} = [\tau_x \; \tau_y \; 0]^T$ about X-and Y-axes and the stiffnesses $k_x(=-\partial F_x/\partial x)$ and $k_y(=-\partial F_y/\partial y)$ of the magnetic trap along X- and Y-axes (Fig. 1b–d). The traces which are parallel to the Y-axis enable actuating in the XZ-plane, while the traces which are parallel to the X-axis enable actuating in the YZ-plane.

If the magnetic field set up by the eight currents is $\mathbf{B} = [B_x \; B_y \; B_z]^T$ then, by virtue of the pitch of the magnet array along X- or

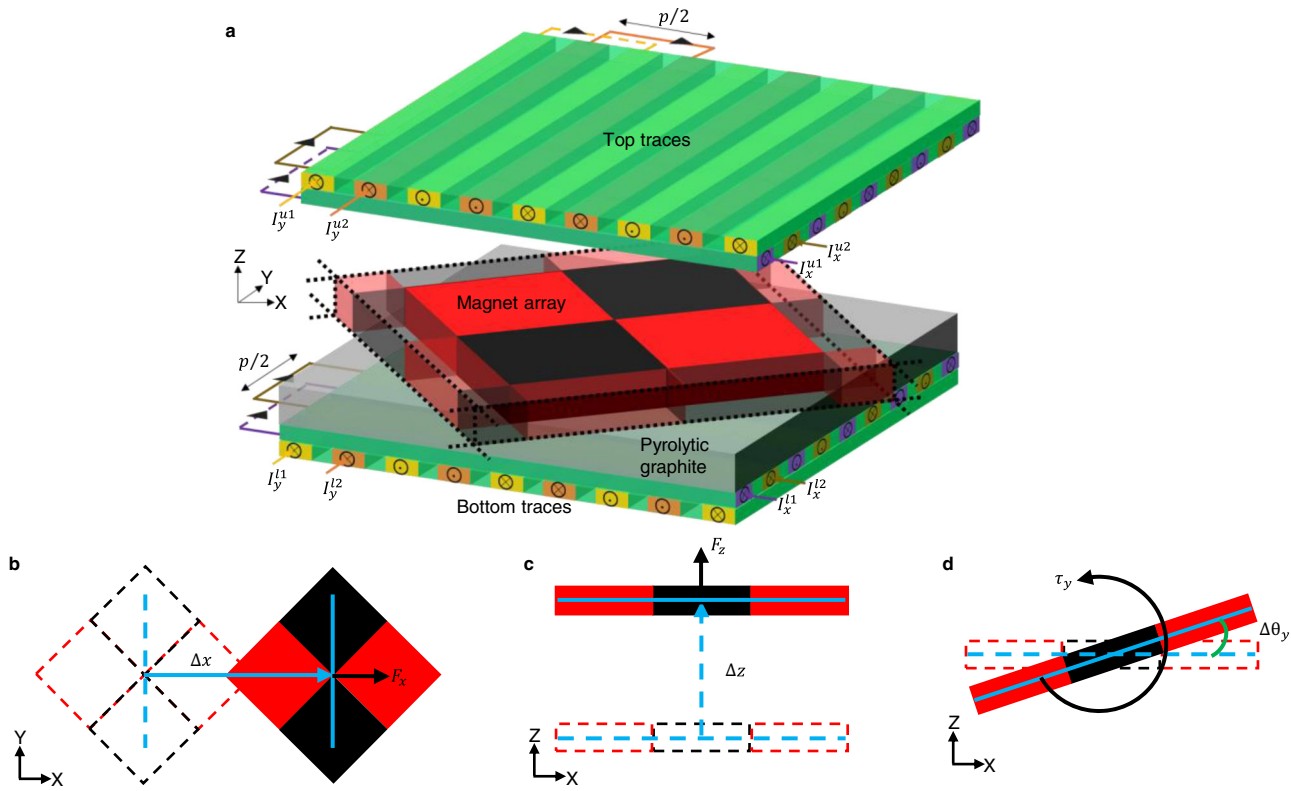

**Fig. 1 Schematics showing the actuator's construction and its motion along different axes. a** The diamagnetically levitated magnetic actuator comprising dual-sided actuation traces, the pyrolytic graphite plate and the magnet array sandwiched between the top and bottom traces. **b** Linear motion $\Delta x$ of the magnet array and the force $F_x$ acting on the array along the X-axis. Similar motion and force can be achieved along the Y-axis as well. **c** Linear motion $\Delta z$ of the magnet array and the force $F_z$ acting on the array along the Z-axis, **d** rotation $\Delta\theta_y$ of the magnet array and the torque $\tau_y$ acting on the array about the Y-axis. Similar rotation and torque can be achieved about the X-axis as well.

Y-axes being the same as that of the traces, magnets with both upward and downward magnetic moments experience identical loads. Thus, if each magnet possesses uniform magnetization of magnitude $M$ and volume $V$, the net force $\mathbf{F}$ on an array of $N$ magnets is given by $\mathbf{F} = N\int_V M\nabla B_z dV$. Likewise, the torque about X- and Y-axes are given by $\boldsymbol{\tau} = N\int_V M\hat{\mathbf{z}}\mathbf{B}\,dV$. The relationship between the forces, torques and the actuating currents can be obtained by exploiting the symmetries in the arrangement of the traces: since all the traces are identical except that they are either offset or rotated with respect to each other, the resulting loads per unit current would also be correspondingly offset or rotated with respect to each other by the same amount as the traces themselves. The overall loads $\mathbf{F}$ and $\boldsymbol{\tau}$ have been obtained following this approach in the supplementary file (see Supplementary Note 2). It is demonstrated that at any point $(x, y)$ in the levitation plane, the overall force $F_x$ and the stiffness $k_x$ are dependent on the linear combination of $I_y^{u1} + I_y^{l1}$ and $I_y^{u2} + I_y^{l2}$. Likewise, the Z-force $F_{zy}$ due to the Y- traces and the torque $\tau_y$ are both shown to depend on the linear combination of $I_y^{l1} - I_y^{u1}$ and $I_y^{l2} - I_y^{u2}$. Similarly, by symmetry, the force $F_y$ and the stiffness $k_y$ are dependent on the linear combination of $I_x^{u1} + I_x^{l1}$ and $I_x^{u2} + I_x^{l2}$, and the force $F_{zx}$ due to the X- traces and the torque $\tau_x$ are both dependent on the linear combination of $I_x^{l1} - I_x^{u1}$ and $I_x^{l2} - I_x^{u2}$. It is worth noting that the force $F_z$ can be generated both by the X- and the Y- traces and the net force is the sum of the two, i.e., $F_z = F_{zx} + F_{zy}$.

Figure 2a–c plot the normalized forces $F_x/F_0$, $F_z/F_0$ and normalized torque $\tau_y/\tau_0$ generated by a single Y- trace as function of the position $x$ of the magnet along the X-axis. Here, $F_0$ and $\tau_0$ represent the characteristic force $F_0$ and characteristic torque $\tau_0$ on a point dipole of magnitude $Nm$ due to a single infinitely long straight segment and are given by $F_0 = \mu_0 NmI/2\pi p^2$ and $\tau_0 = \mu_0 NmI/2\pi p$. The figures show that by virtue of $p \sim z_0$, all the loads vary nearly sinusoidally with $x$ and the first harmonics of normalized loads $F_{x1}/F_0$, $F_{z1}/F_0$ and normalized torque $\tau_{y1}/\tau_0$ are nearly of the same amplitude as the actual waveform. Under the approximation that the variation in magnetic field along the X- and Y-axes are sinusoidal, the expressions for the loads experienced by the magnet can be simplified. In particular, an array of $N$ square magnets, each with magnetic moment of magnitude $m$, can be replaced by a single point dipole $m'$ at their geometric center, given by (see Supplementary Note 2.3)

$$m' = 4Nm/\pi^2 \qquad (1)$$

Furthermore, let $b_x$ and $b_z$ represent the X- and Z-components of magnetic field per unit current set up in the levitating plane by a single trace aligned along the Y-axis while $b_{1x}$ and $b_{1z}$ are the amplitudes of the first harmonic of $b_x$ and $b_z$. Defining vectors $\mathbf{I_y}$ and the normalized load $\mathbf{F_{Iy}}$ as $\mathbf{I_y} = \begin{bmatrix} I_y^{l2} - I_y^{u2} & I_y^{l1} - I_y^{u1} & I_y^{l1} + I_y^{u1} & I_y^{l2} + I_y^{u2} \end{bmatrix}^T$ and $\mathbf{F_{Iy}} = \frac{1}{m'} \begin{bmatrix} -\frac{p}{b_{1x}2\pi}F_{zy} & \frac{1}{b_{1x}}\tau_y & \frac{p}{b_{1zy}2\pi}F_x & -\frac{p^2}{b_{1zy}4\pi^2}k_x \end{bmatrix}^T$, it can be shown that $\mathbf{I_y}$ is related to $\mathbf{F_{Iy}}$ as (see Supplementary Note 2.3)

$$\mathbf{I_y} = \begin{bmatrix} \mathbf{R}(-2\pi x/p) & \mathbf{0}_{2\times 2} \\ \mathbf{0}_{2\times 2} & \mathbf{R}(2\pi x/p) \end{bmatrix} \mathbf{F_{Iy}} \qquad (2)$$

where $\mathbf{R}(2\pi x/p)$ is a rotation matrix given by $\mathbf{R}(2\pi x/p) =$

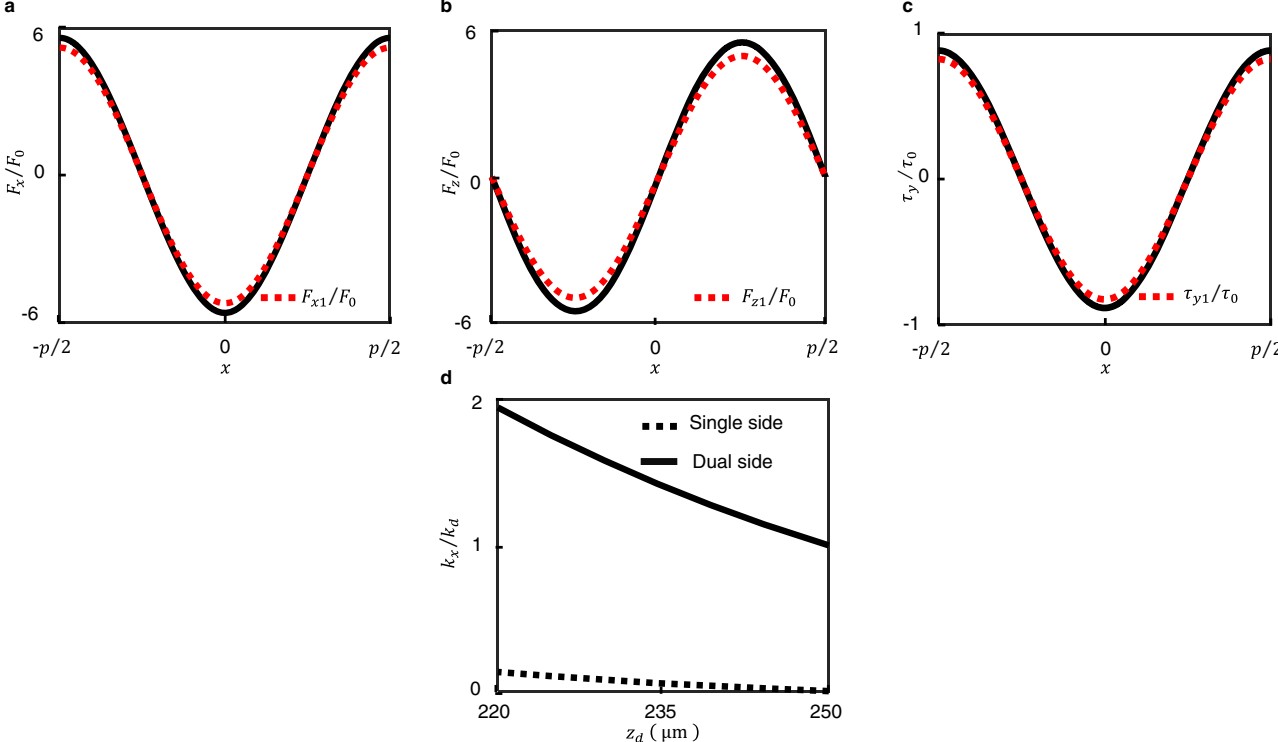

**Fig. 2 Plots showing the variation of loads along the X-axis and the stiffness along the Z-axis.** Plots showing the normalized loads due to a single trace acting on the magnet array as function of the position $x$ of the magnet along the X-axis: (**a**) X-force $F_x/F_0$, (**b**) Z-force $F_z/F_0$, (**c**) Y-torque $\tau_y/\tau_0$. The plots shown in dotted red lines in **a**–**c** represent the first harmonic of the periodic waveforms represented by solid black lines. **d** Plot showing the dependence of the normalized X-stiffness $k_x/k_d$ on the Z-position of center of the magnet array above the graphite plate for the cases of single-sided and dual-sided actuation. In the calculations, the thickness of the magnet was considered as 0.4 mm, for which case, the levitation height is $z_d = 251\,\mu m$ without any actuation. The diamagnetic stiffness $k_d$ at a levitation height of $z_d = 251\,\mu m$ was obtained from finite element analysis to be 0.94 N/m.

$\begin{bmatrix} \cos(2\pi x/p) & -\sin(2\pi x/p) \\ \sin(2\pi x/p) & \cos(2\pi x/p) \end{bmatrix}$. Similarly, defining $\mathbf{I_x}$ and $\mathbf{F_{Ix}}$ as $\mathbf{I_x} = \begin{bmatrix} I_x^{l2} - I_x^{u2} & I_x^{l1} - I_x^{u1} & I_x^{l1} + I_x^{u1} & I_x^{l2} + I_x^{u2} \end{bmatrix}^T$ and $\mathbf{F_{Ix}} = \frac{1}{m'} \begin{bmatrix} -\frac{p}{b_{1x}2\pi}F_{zx} & \frac{1}{b_{1x}}\tau_x & \frac{p}{b_{1zy}2\pi}F_y & -\frac{p^2}{b_{1zy}4\pi^2}k_y \end{bmatrix}^T$, it can be shown that

$$\mathbf{I_x} = \begin{bmatrix} \mathbf{R}(-2\pi x/p) & \mathbf{0_{2\times2}} \\ \mathbf{0_{2\times2}} & \mathbf{R}(2\pi x/p) \end{bmatrix} \mathbf{F_{Ix}} \quad (3)$$

Equations (2) and (3) can be employed to generate the necessary elements of $\mathbf{I_y}$ and $\mathbf{I_x}$ to ensure that at any point $(x,y)$ in the levitation plane, it is possible to apply any desired load $\mathbf{F}$, $\tau$ and achieve stiffnesses $k_x$ and $k_y$. For a desired Z-force $F_z$, the corresponding force $F_{zx}$ can be arbitrarily chosen, and $F_{zy}$ would be $F_z - F_{zx}$. The maximum loads that can be applied are ultimately limited by the magnitude of the maximum current $I_{max}$ that can be passed through the traces, which in turn, is chiefly limited by Joule heating caused by passing higher currents. The corresponding maximum forces along X-, Y- and Z-axes are given to be $F_{x,max} = F_{y,max} = (4\sqrt{2}\pi m' b_{1zy}(z_0)/p)I_{max}$, $F_{z,max} = (8\sqrt{2}\pi m' b_{1x}(z_0)/p)I_{max}$. The maximum torque is given to be $\tau_{x,max} = \tau_{y,max} = (8\sqrt{2}m' b_{1zy}(z_0)/\pi^2)I_{max}$. (see Supplementary Note 2.4).

While the forces on a free magnet array in the X–Y plane are entirely due to the traces, along the Z-axis, the array also experiences the diamagnetic repulsion force $F_d$ and the downward weight, in addition to the electromagnetic force $F_z$. Thus, for an unconstrained magnet array in static equilibrium, the stiffness of the trap along the Z-axis is given by $k_z = -\partial F_z/\partial z - \partial F_d/\partial z$.

Maxwell's equations reveal that $k_x + k_y - \partial F_z/\partial z = 0$ (see Supplementary Note 2.5). Therefore, for an unconstrained array, the stiffnesses along X-, Y- and Z-axes are related as $k_x + k_y + k_z = -\partial F_d/\partial z = k_d(z_d)$. Thus, though $k_x$ and $k_y$ can be chosen arbitrarily, for the trap to be stable along all axes, it is necessary that each of them be less than $k_d(z_d)$. For the case when the stiffness along X-, Y- and Z-axes are identical, the maximum achievable stiffness along each axis is $k_d(z_d)/3$. In contrast, for the case of single-sided actuation, there exists cross-coupling between $k_x$, $k_y$ and $F_z$. Thus, any attempt to increase the in-plane stiffnesses also reduces the gap between the magnet and graphite, and the upper limit to achievable stiffness is decided by the current at which the magnet array makes contact with graphite. Figure 2d plots the maximum achievable stiffness for single-sided actuation and compares the same with dual-sided actuation. All the stiffnesses have been normalized with respect to $k_d(z_d)$. It is seen that the achievable in-plane stiffness for single-sided actuation starts from zero in the levitation plane, and gradually increases as the gap is reduced. However, it remains substantially lower than that of dual-sided actuation at all levitation heights. For dual-sided actuation, the stiffness even in the levitation plane is about 5 times higher than the maximum stiffness achieved with single-sided actuation. It is also worth noting that by employing another graphite layer beneath the top PCB, further improvements in stiffness can be achieved for dual-sided actuation.

An important metric of the actuator is its workspace. The area of the workspace along X- and Y-axes is determined by the area of the PCBs covered by the meanders along each axis. The workspace along the Z-axis is limited from below by the

possibility of contact with the graphite plate, while it is limited from above by the height at which the magnet array becomes unstable. The workspace below the levitation plane is equal to the air gap, i.e., $z_0 - t_d - t/2$, where $t$ represents the thickness of the magnet array. Above the levitation plane, it is limited by the height $z_1$ at which $k_z = 0$ (see Supplementary Note 2.6), i.e., when $k_d(z_1) = k_x + k_y$. Thus, the workspace along the Z-axis is $z_1 - t_d - t/2$. This is typically much smaller than $z_0$ since the air gap between the magnet array and the graphite plate is small, and $k_d(z)$ reduces rapidly for displacements above the graphite plate. Thus, within the workspace, the variation of $k_x$, $k_y$ along the Z-axis would be small and would not affect most practical applications of the actuator. However, the Z-stiffness can still change substantially within the workspace, due to the strong dependence of $k_d(z)$, and hence $k_z$, on the height above the graphite plate. Employing another graphite plate beneath the top PCB helps to substantially reduce the variation in the Z-stiffness within the workspace and would be suitable for applications that require small motion range along the Z-axis.

Due to the significantly higher stiffnesses, dual-sided actuation enables achieving higher vibration immunity, while the fact that the magnet array is sandwiched between the two PCBs also provides it with greater immunity from acoustic noise. The ability to apply independent forces and moments of large magnitudes has several implications: first, it implies that the actuator can carry large payloads. Second, it implies that the actuator can be coupled to a greater diversity of loads. Third, in combination with position feedback, it can even drive compliant loads of large stiffness. Finally, the large forces can also be employed to achieve fast motion of the magnet array. However, with the proposed configuration of actuating traces, rotation of the magnet array about the Z-axis is not possible. Furthermore, due to the limited gap $2z_0$ between the two PCBs, the motion ranges are large only for translation along the X- and Y-axes. The first lacuna can be addressed either by incorporating two or more zones of actuating traces. The second can be addressed linking two or more actuators through mechanical interconnections. In the first case, multiple zones of actuating traces are patterned on the PCBs, with each zone comprising the same four meandering traces described earlier, but which are controlled independent of each other. Figure 3a illustrates this strategy, wherein a single rigid magnet array is arranged in the form of a cross, with each arm of the cross actuated by a separate zone of traces. In literature, such zone-based actuation has been employed to independently actuate multiple levitating magnet arrays[47,48]. Here, the multi-zone positioner is employed to apply equal but opposite forces on the diagonal arms of a single array and thereby generate a couple, which in turn, rotates the array about the Z-axis as shown in Fig. 3b. In practice the forces were applied by shifting the equilibrium points in diagonally opposite zones by equal and opposite amounts, so that the array rotates to its new orientation.

The multi-zone nano-positioning system also enables simultaneous positioning along all the six degrees-of-freedom. To obtain the drive configuration, i.e., the currents necessary for simultaneous multi-degree-of-freedom positioning, it is first noted that within the pitch $p$ of a single meander, there is a unique relationship between the loads and the actuation currents and that this repeats itself with a periodicity of $p$ along the X- and Y-axes. This fact is employed to obtain the drive configurations to achieve the desired multi-degree of freedom position in three steps: In the first step, the specified X- and Y-displacements and rotation about the Z-axis are achieved. In the second step, the loads that need to be applied to achieve the specified Z-position, and rotation about X- and Y-axes are obtained. In the third step, the X- and Y-stiffnesses are also specified and the actuation currents necessary to apply the loads and achieve the specified

stiffnesses at the specified X-, Y-position are obtained using Eqs. (2) and (3). The resulting relationship between the necessary currents and the specified displacements are described in Supplementary Note 3.1.

The limited out-of-plane motion range can be addressed by linking two or more actuators with a compliant mechanism. An example realization is shown in Fig. 3c, wherein a symmetric trapezoid-shaped mechanism, with compliant hinges at each corner of the trapezoid, is actuated on either side by two actuators. The sides of the trapezoid are of length $\ell$ and are initially tilted at an angle $\theta_0$. This arrangement too provides six-degrees-of-freedom for the central platform, with the added feature of a large Z-range. To move the platform along the Z-axis, equal but opposite forces are generated in the two actuators (Fig. 3d). The resulting displacement $\Delta x$ changes the angle of tilt of the sides to $\theta(x) = \cos^{-1}[\cos\theta_0 - \triangle x/\ell]$ and displaces the stage by the amount $\triangle z(x) = \ell\sin\theta - \ell\sin\theta_0$. Thus, the range of $\Delta z$ is decided by $\ell$, which can be much bigger than $z_0$. It is worth noting that it is possible to also rotate the platform about the Y-axis by large angles by displacing the two actuators appropriately. In applications where large motion range is desired along the Z-axis, it is preferable to employ this actuation strategy since the motion of the platform can be restricted to a single plane, viz., the levitation plane. Thus, the variation of stiffnesses with Z-position will not be a matter of concern.

**Development and characterization of the positioning stages**. In the experimental set-up realized in the laboratory, each square-shaped magnet in the magnet array was made from an alloy of Neodymium, Iron and Boron and possessed an edge of length 1.7 mm and thickness 400 μm. They were levitated above a graphite plate of thickness $t_d = 500$ μm. The resulting levitation height of the center of the magnets above graphite was found to be $z_d = 251$ μm and thus, the gap between the magnets and the graphite was 51 μm. Based on the size of the magnets, the pitch $p$ of the traces was chosen to be 2540 μm. A 3D micrometer stage was employed to adjust the position of the top traces and ensure that the top and bottom traces were aligned. The actuators were then positioned below a microscope to view in-plane motion. A side microscope was employed to view the out-of-plane motion. The images were acquired using Complementary Metal Oxide Semiconductor (CMOS) cameras and sub-pixel digital image correlation (DIC) was employed to measure the fine motion in both cases. The measurement was acquired with a maximum image acquisition rate of 250 Hz and actuation was performed by a real-time controller (DS1104) which was operated at 10 kHz update rate. The entire experimental set-up including the positioner and the microscopes were mounted on a vibration isolation platform (HOLMARC, PVISA 180-120) to reduce the effect of external vibration on the measurement and positioning accuracy.

Figure 4a shows the multi-zone positioner while Fig. 4b, c shows the top view and the side view of the cross-shaped magnet array alone. The center of the magnet array was employed as a positioning stage to first validate the improvements in stability, precision and cross-axis motion achievable with dual-sided actuation in comparison with single-sided actuation. Figure 4d compares the measured in-plane vibration of a trapped magnet array for single-sided actuation with that of dual-sided actuation. The RMS noise in position for single-sided actuation is 176 nm. In contrast, the RMS noise for the dual-sided positioner is 1.88 nm. It is worth noting that the measurement system has a noise floor of 1.85 nm. Thus, under the assumption that measurement noise and stage vibration are uncorrelated, the estimated RMS vibration of the array is about 0.11 nm. The improvement is attributed primarily to higher stiffness and the

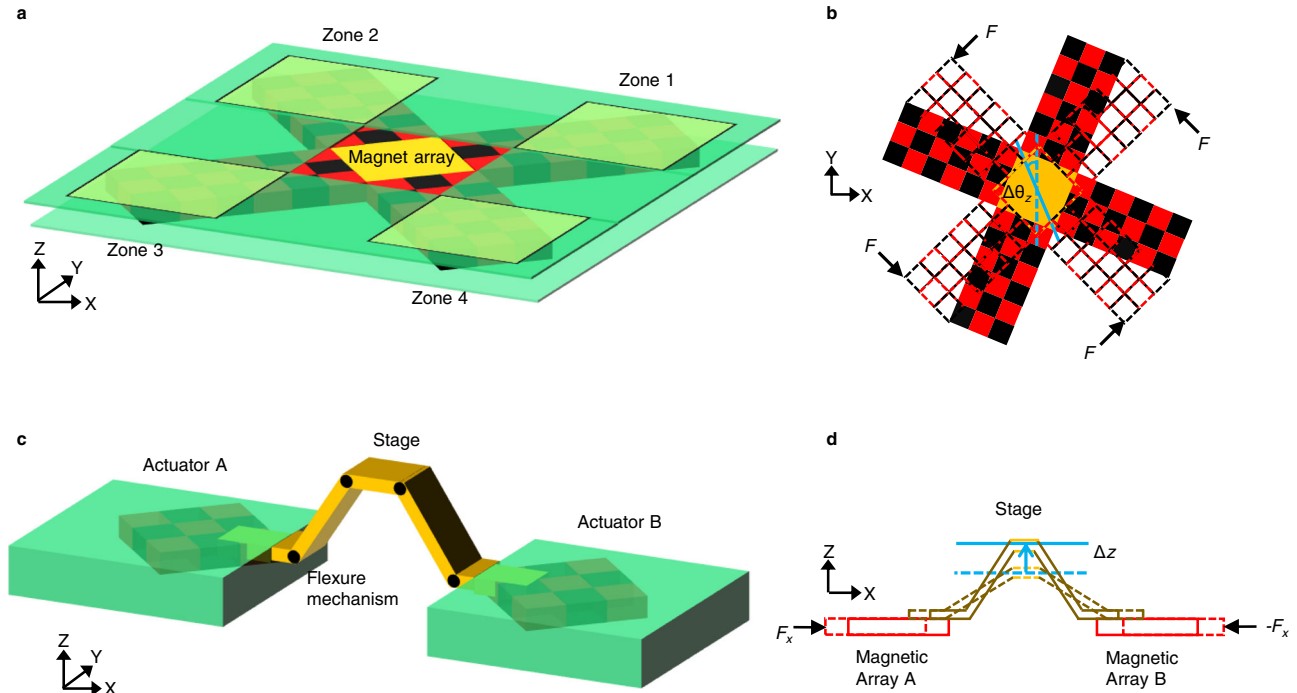

**Fig. 3 Schematics of the multi-zone and flexure-mechanism based positioners. a** The multi-zone positioner wherein each of the four arms of the cross-shaped magnet array are actuated by four different zones on the PCBs. **b** Rotation of the cross-shaped magnet array about the Z-axis by large angles $\Delta\theta_z$ by applying equal but opposite horizontal forces $F$ to the diagonal arms of the positioner. **c** The flexure-based positioner obtained by integrating a compliant 5-bar trapezoidal mechanism between two actuators. **d** Large-range linear displacement $\Delta z$ of the central stage along the Z-axis achieved by applying equal and opposite forces $F_x$ on the actuators along the X-axis.

associated higher natural frequency with dual-sided actuation. Also, sandwiching the magnet array between two PCBs reduces the area of the array exposed to the effect of acoustic noise. Figure 4e showcases the positioning precision of dual-sided actuation, by executing 50 nm-sized stepping motion along the X-axis. Since the observed noise is primarily measurement noise, a filtered version of the same staircase is also plotted, since this is expected to resemble the actual motion of the array. Figure 4f compares the cross-axis Z-motion as the magnet array was translated along the X-axis. It is seen that while single-sided actuation not only shifts down the levitation plane by about 24 μm, the peak-to-peak change is about 11 μm. In contrast, dual-sided actuation leads to negligible change in the levitation height and the peak-to-peak variation is about 2 μm. This small change can also be eliminated by calibrating the actuation gain experimentally and use it to compensate for cross-axis motion. Finally, to verify the ability to carry higher payloads, an object whose weight was 16% of the weight of the magnet array was placed on the central platform. The payload was placed exactly in the center of the magnet array to avoid any tilt due to the uneven distribution of the weight of the payload. This initially resulted in a small reduction in levitation height, but by gradually increasing $I_y^{u2} - I_y^{l2}$, the array could once again be levitated at its original height (Fig. 4g). With higher difference $I_y^{u2} - I_y^{l2}$, higher payloads can be carried. For the fabricated PCB, it is possible to pass $I_{max} = 2$ A through the traces without significant overheating. For this case, maximum in-plane force is about 13 times the overall weight of the overall magnet array (see Supplementary Note 2.4), while the maximum out-of-plane force is about 27 times the weight of the overall array. It is also worth noting that the maximum-in plane force for dual-sided actuation is ~25 times greater than the maximum in-plane force for single sided-actuation, while the maximum payload carrying capacity for the

dual-sided actuation is ~50 times greater than the maximum payload carrying capacity for the single-sided actuation (see Supplementary Note 2.4). The workspace of the positioner below the levitation plane was determined to be 51 μm while above the levitation plane it was determined to be 50 μm for the case $k_x = k_y = k_d/3$. The percentage change in X- and Y-stiffness was about 1.4% within the workspace, which is small and not significant for most practical applications. The currents necessary to position the actuator anywhere in this workspace can be obtained from the equations for the drive configuration. Numerically, the currents were found to be about 245 mA, which is much lesser than the maximum current $I_{max}$. This verifies that for the chosen parameters, the current limits do not influence the workspace of the device.

The multi-zone positioner was subsequently tested for its ability to move along all the six degrees-of-freedom (see Supplementary Movie 1). The currents were generated using Eqs. (2) and (3), where all the loads were set to zero, and the stiffnesses $k_x$, $k_y$ were chosen to be a constant, while the displacement $\Delta x$ was changed at a constant rate $v$ with respect to time, i.e., $\Delta x(t) = vt$. Figure 5a shows the displacement of the stage $\Delta x(t)$. The stage is seen to smoothly follow the commanded change in position over a range of about 5 mm, with good linearity. It is also seen from the forward and reverse waveforms that hysteresis is absent, and the motion is repeatable. The small undulations is attributed to the approximations involved in obtaining Eqs. (2) and (3) and can be eliminated by calibration and compensation. Similar motion was achieved even along the Y-axis and has not been separately shown on account of symmetry.

It is worth noting that along the X-, Y- axes the motion range is independent of both the stiffness and the speed of response and depends only on the extent of meanders on the PCBs along these axes. Along the Z-axis, there exists a trade-off between the size of

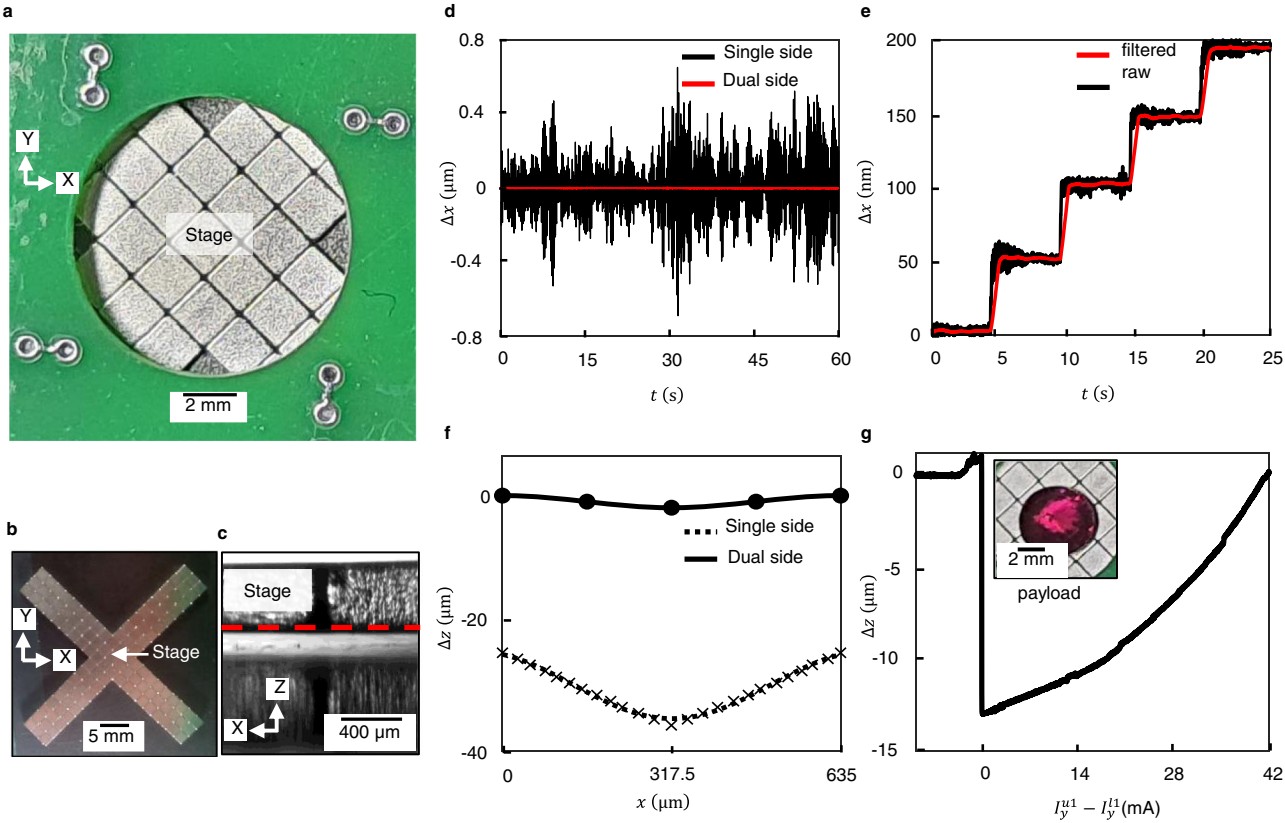

**Fig. 4 Photographs of the multi-zone positioner and plots characterizing its positioning performance. a** Photograph showing the multi-zone, dual-sided positioner. **b** Photograph showing the top-view of the cross-shaped magnet array, whose center acts as a positioning stage, along with its four magnetic arms. **c** Micrograph showing the side view of the levitating magnet array. **d** Plots comparing the motion $\Delta x$ of the magnet array for single-sided and dual-sided actuation with respect to time $t$. The current through all the traces were maintained constant at 250 mA for dual-sided actuation and at 40 mA for single-sided actuation. Any further increase in the current in the latter resulted in contact of the magnet array with the graphite plate. **e** Plot showing the 50 nm stepping of the magnet array along the X-axis with respect to time $t$ for dual-sided actuation. The current through one of the Y-traces in each zone was increased in steps of 110 µA while the remaining track currents were kept constant. **f** Plots comparing the cross-axis motion $\Delta z$ of the magnet array as a function of its X-position, $x$ for the single- and dual-sided actuation. **g** Plot showing 14 µm reduction in the levitation height of the magnet array due to addition of the payload of weight 1.8 mN and subsequent restoration of the levitation height by increasing $I_y^{u1} - I_y^{l1}$, i.e., the difference between top and bottom trace currents.

the workspace and stiffness, wherein higher X- and Y-stiffnesses leads to reduction in workspace along the Z-axis. Figure 5b shows translation $\Delta z(t)$ within the workspace along the Z-axis achieved by controlling $I_y^{u1} - I_y^{l1}$. The nonlinearity in the motion is due to the nonlinear dependence of the equilibrium position along the Z-axis on $I_y^{u1} - I_y^{l1}$. Figure 5c shows the rotation of the stage by $\Delta\theta_z$ about the Z-axis achieved by generating equal but opposite in-plane forces in the diagonally opposite arms of the array. The motion is smooth with range of $\pm31.5°$. The inset shows fine angular positioning and demonstrates the RMS noise to be about 50 µrad. Figure 5d shows rotation about X-axis, achieved by applying upward force to two of the arms and downward force to the two other arms, with the resulting rotation of the stage $\Delta\theta_y$ being $\pm0.2°$. Similar motion was achieved about the X-axis and has not been shown on account of symmetry. Thus, Fig. 5a–d showcase the ability to move along all six degrees-of-freedom in a smooth, non-hysteretic manner. It is worth noting that the ability to displace of the magnet array along any axis also implicitly demonstrates the ability to apply load on the magnet array along that axis. From the view-point of dynamics, the magnet array behaves as a rigid body that is electromagnetically trapped with the trap stiffness being $k_x$, $k_y$ and $k_z$ along X-, Y- and Z- axes

respectively. Thus, it can be modeled as a mass-spring-damper system, where the damping arises due to the surrounding air. This model was validated experimentally and the bandwidth of the nanopositioner along the linear channels was evaluated from the step responses along each axis. The open-loop bandwidth along X-, Y- and Z-axes were $\omega_{bx} = \omega_{by} = 131$ rad/s, $\omega_{bz} = 153$ rad/s, while the bandwidth for rotations about X-, Y- and Z-axes were $\omega_{b\theta_x} = \omega_{b\theta_y} = 121$ rad/s and $\omega_{b\theta_z} = 90$ rad/s (see Supplementary Note 3.2).

To showcase the large out-of-plane motion range achievable by connecting the actuators to the complaint stage (see Supplementary Movie 2), the compliant element was chosen to be a strip of paper of thickness 150 µm that was bent into the form of a trapezium with sides of length l=1.4 cm and base angle $\theta_0 = 17°$ (Fig. 6a). The two actuators were simultaneously moved towards each other by $\Delta x = 1.2$ mm with uniform velocity. The resulting out-of-plane displacement $\Delta z(t)$ is plotted in Fig. 6b. The out-of-plane displacement of $\Delta z = 900$ µm represents an 18-fold improvement over the maximum range achievable with single-sided actuation.

**Application: a compact in-situ tip-replacement module.** One key benefit of the reported nano-positioning stages is the high

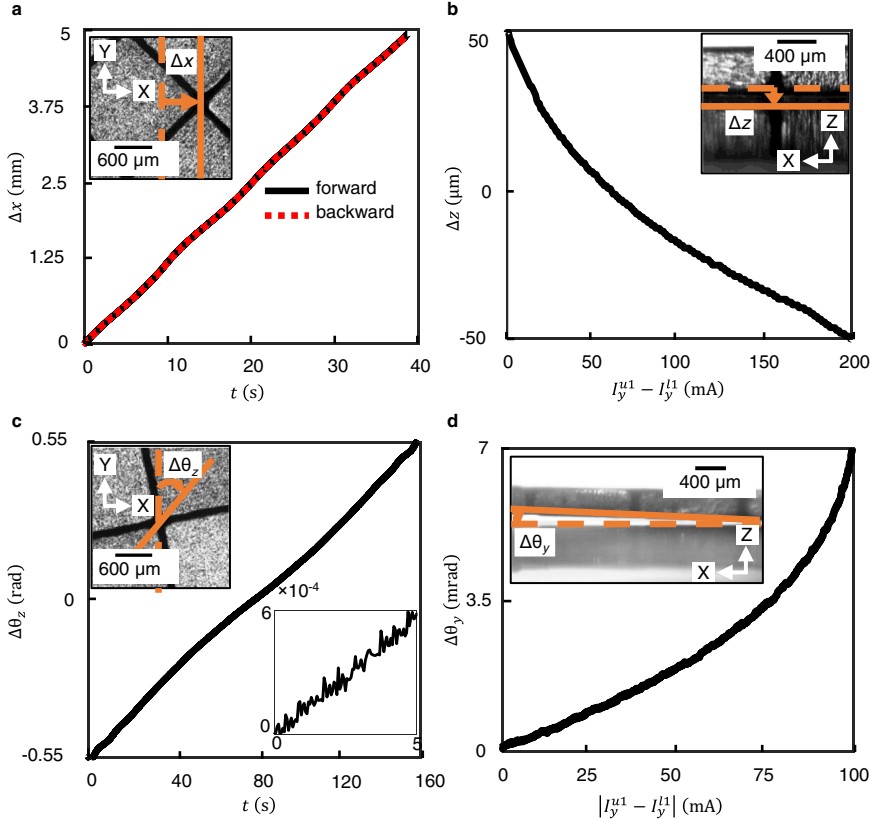

**Fig. 5 Plots showing the linear and angular motion characteristics of the multi-zone positioner. a** Backward and forward motion $\Delta x$ of the multi-zone positioner along the X-axis traversing 5 mm. The X-position of the trap was changed with uniform velocity of $v = 0.125$ mm/s. **b** Linear motion $\Delta z$ of the multi-zone positioner along the Z-axis traversing 100 μm. **c** Rotational motion of the multi-zone positioner along the Z-axis traversing ±31.5°. The inset shows the close-up view and demonstrates the resolution to be 50 μrad (**d**) rotational motion of the six-axis positioner along the Y-axis traversing 0.4°. The trapping stiffness along the X- and Y-axes were chosen to be $k_x = k_y = 0.2$ N/m, which was achieved by choosing the amplitude of current to be 200 mA through both the X- and Y-traces.

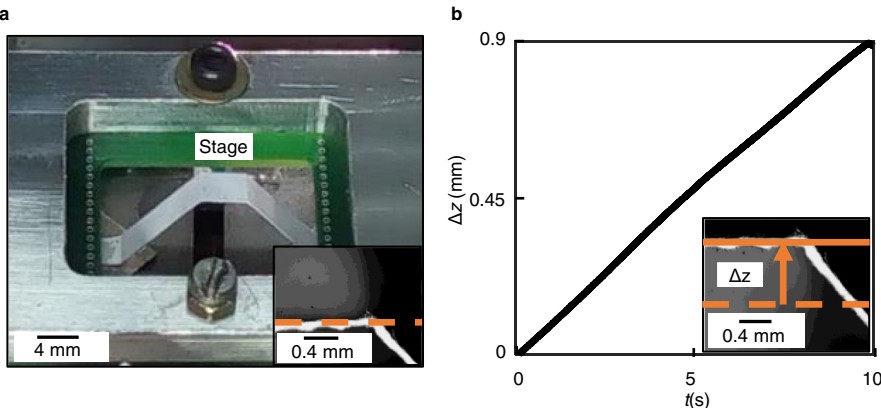

**Fig. 6 Photograph of the flexure-based positioner and plot showing its motion characteristics along the Z-axis. a** Photograph of the flexure-based positioner showing the compliant 5-bar trapezoidal mechanism integrated with the actuators. Each of the magnet array comprises of 64 magnets arranged as an 8 × 8 matrix. **b** Plot showing the linear motion $\Delta z$ of the compliant stage with respect to time $t$ along the Z-axis traversing 0.9 mm.

degree of compactness with which multi-degree of freedom actuation is achieved. This makes them especially suited for applications that require precision positioning in a confined space, for example, in retrofitting existing instruments with a compact multi-degree-of-freedom actuator module.

To showcase this capability, these stages have been employed here for in-situ tip-replacement in an Atomic Force Microscope (AFM). AFM is a widely used instrument for nanometer-scale imaging, characterization and manipulation[49–51]. Tip-replacement is an unavoidable aspect of AFM in every application due to the frequent blunting and contamination of the tip of an AFM cantilever[52]. While tip-replacement is conventionally accomplished by replacing the entire probe, replacing just the tip of the probe has a range of benefits, including reduced cost and

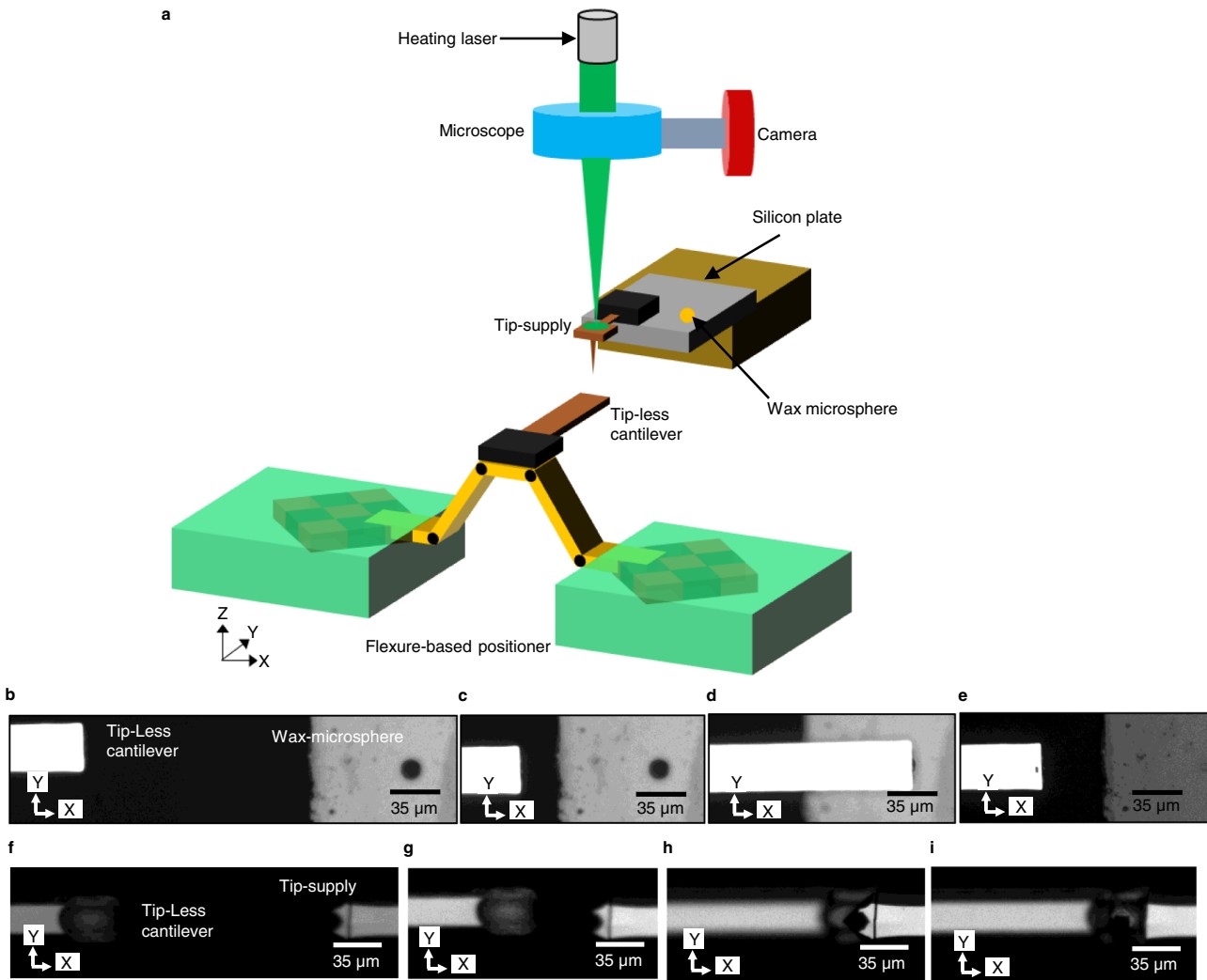

**Fig. 7 Schematic of the experimental set-up and micrographs acquired during automated tip pick-up. a** Schematic of the set-up used for performing tip-replacement of an AFM cantilever. Optical micrographs showing (**b**) the probe and wax within the same field of view, (**c**) coarse positioning of the probe, (**d**) fine positioning and docking of the tip-less probe on the wax microsphere, (**e**) wax picked-up by the probe, (**f**) the probe with molten wax and the tip-supply within the same field of view, (**g**) coarse positioning of the probe, (**h**) fine positioning and docking of the probe on the tip-supply, (**i**) picked-up tip. The cantilever employed in the experiment had length, width and thickness of 400 μm, 35 μm and 2.5 μm respectively. The diameter of the paraffin wax microsphere used in the experiment is 15 μm.

greater versatility. Previously tip-replacement has been performed by employing standalone hardware, which is therefore unsuitable for replacing tips in-situ[53,54]. Here the flexure-based positioner has been employed as a compact tip-replacement module. The procedure for tip-replacement has been developed in-house and is based on attaching a new tip to the probe using paraffin wax as a bridging layer, and the tip is detached by photo-thermally melting the wax microspheres[55]. The central part of this module is the flexure-based positioning stage reported here on which a tip-exchange platform is mounted. The schematic of the set-up employed for performing tip-replacement is shown in Fig. 7a. The steps involved include docking a tip-less AFM probe on a wax micro-sphere by performing coarse and fine positioning in 3D, picking up the microsphere, and then docking the probe on a detachable tip-head, again by the aid of coarse and fine positioning, forming a wax bridge between the tip-head and the probe and then employing the tip-head stuck to the probe to perform imaging. While conventionally, both coarse and fine positioning needed to be done by two separate stages, namely by a motorized micrometer stage and a piezo-actuated stage, here both

operations are done by the 3-axis positioner. This reduced the space required for the positioners from about four thousand cubic centimeters down to a few tens of cubic centimeters. The process of tip-pickup using the reported flexure-based positioner is showcased in Fig. 7b–i.

## Discussion

This paper proposed dual-sided actuation for diamagnetically levitated magnet array. The actuation traces were patterned on a PCB and together carry 8 independent currents, which were demonstrated to independently control 3D forces on the magnet array, the torques about X- and Y-axes, and the stiffnesses about X- and Y-axes. The approximately sinusoidal variation of magnetic field in the levitation plane enabled replacing the magnet array with an equivalent point dipole, and obtaining simple expressions relating the currents to the loads and stiffness. It was also shown that multi-zone positioners can achieve six degrees-of-freedom motion with large rotation range about Z-axis while compliant mechanisms can be employed to connect

two actuators and transform the large in-plane motion range to large range along the Z-axis. Experimentally, the positioners were shown to possess positioning precision better than 1.88 nm, substantially reduced cross-axis motion and the ability to carry payloads without displacing from the levitation plane. The positioner employing multi-zonal actuation was shown to displace along all six degrees-of-freedom, with range of 5 mm for in-plane displacements and ±31.5° for rotation about the Z-axis. Likewise, a flexure-based stage, obtained by connecting the actuators to a compliant trapezoidal mechanism, enabled achieving motion range of about 900 µm along the Z-axis. Finally, to showcase utility, the flexure-based stage was employed in the context of tip replacement in AFM to replace bulky positioners and bring the volume down to just a few cubic centimeters. In comparison to other diamagnetically levitated stages, the design achieves over 100-fold improvement in precision, 18-times larger Z-range and 3-times larger rotation range. In comparison with active levitation stages, the design is over one order of magnitude lesser in volume, and over two orders of magnitude higher in angular range. In comparison with piezo-actuator based stages, the proposed design achieves comparable positioning precision but with over one order of magnitude larger range and with about an order of magnitude smaller volume. Thus, the proposed stages can be employed for a variety of nano-positioning tasks, especially those that require both large range and high-precision positioning and is especially suited for integration with other instruments with complementary capabilities.

## Methods

**Development of the actuator and the six axis positioners**. The magnet array was a chequerboard of permanent magnets (alloy of Neodymium, Iron and Boron, of grade 52), each with a square face of side 1.7 mm and thickness 0.4 mm. The pitch of the array was hence $p = 2.54$ mm. The PCB was made in two layers and traces on the PCBs were 70 µm thick and 254 µm wide. The traces in X-direction were 200 µm below the traces in the Y-direction, while the two traces in any direction were co-planar, and comprised 48 straight segments, each of length 30 mm. A pyrolytic graphite block was fixed to the bottom PCB and was milled down to a thickness of 500 µm. In the final step the surface was lapped to ensure flatness. The magnet array levitated at a height of $z_d = 250$ µm above the graphite block and therefore at a height of $z_0 = 750$ µm above the bottom PCB. The top PCB was positioned such that the gap between the two PCBs was $2z_0$.

The X-shaped magnet array in the multi-zone positioner comprised a total of 117 magnets, with each arm of the array comprising of 27 magnets and the central platform comprising of 9 magnets.

In the case of the flexure-based positioner, each actuator employed an $8 \times 8$ array of magnets. The compliant stage was made by employing a rectangular piece of paper (elastic modulus ~2 GPa) of thickness 150 µm, overall length 50 mm and width 10 mm, which was bent into the shape of a trapezoid of the slanting side length being l = 14 mm and tilt angle $\theta_0 = 17°$.

**Actuation and motion characterization**. The currents through the PCBs were controlled by a real-time controller (DS1104, dSPACE) in combination with the necessary current drivers, and the controllers were developed using MATLAB$^R$ SIMULINK$^{TM}$ software and was operated at 10 kHz sampling rate. The motion of the positioning stage was characterized by a modular top microscope (BXFM, Olympus) and a modular side microscope (Cerna, Thorlabs) each attached with a Complementary Metal Oxide Semiconductor (CMOS) camera (MC050MG, Ximea), which could be used to measure displacements at a maximum rate of 250 frames per second. The top microscope employed two different objective lenses of magnification 2.5 (MPLFLN2.5x, Olympus) and 50 (LMPLFLN50x, Olympus), with the former used for characterization of range and the latter for resolution of the positioner respectively. The tube lens employed with the top microscope had a magnification of 0.63 (U-TV0.63XC, Olympus). The side microscope employed a long working distance objective lens of magnification 5 (M Plan APO 5x, Mitutoyo) and a tube lens of magnification 0.75 (Thorlabs) for characterizing the range of the positioner along the out-of-plane axis. To extract the displacements, the acquired images were analyzed using sub-pixel Digital Image Correlation that employed the Newton-Raphson (N-R) algorithm. The algorithm was written in MATLAB software. The N-R algorithm enabled measuring displacements with 0.01 pixel resolution. The bottom and top PCBs were both placed on compact micrometer stages so that the traces on them could be precisely aligned.

**Development of the tip-exchange platform**. The tip-exchange platform comprises wax-microspheres and a tip-supply positioned on a silicon substrate. Wax microspheres were generated using the centrifugal atomization technique, wherein molten paraffin wax was poured on a heated aluminum disk that was rotating at 3600 rotations per minute. The disk was maintained at 75 °C, i.e., above the melting point of the wax, by means of current-carrying chromium wire which was positioned below the rotating plate. The outward motion of the molten wax due to centrifugal forces resulted in the generation of wax microspheres, which were collected on a glass slide. A glass micropipette was employed to pick-up a wax microsphere of diameter about 15 µm and subsequently place it at the desired position on the substrate besides the tip-supply. The tip-supply was fabricated by using Focused Ion Beam milling (Helios Nano Lab 600i, FEI) to mill a thin neck on an AFM cantilever of length, width and thickness 1 µm, 0.4 µm and 0.3 µm. The length, width and thickness of the original cantilever was 210 µm, 30 µm and 2.7 µm respectively (All-In-One Cantilever B, Budget Sensors).

## Data availability

The data that supports the finding of this work are available from the corresponding author upon reasonable request.

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

## Acknowledgements

This material is based on work supported by the Department of Science and Technology, India under Grant no. DST1565 and Imprint under Grant no. IMPR0006. Any opinions, finding, conclusions, or recommendations expressed in this material are those of the authors and do not necessarily reflect the views of the Department of Science and Technology and Imprint. The authors wish to thank Ms. Pranali Shriram Gaydhane for assistance in developing the code for position measurement.

## Author contributions

K.S.V. and G.R.J. conceived the idea and performed the analyses. K.S.V. performed the experiments and processed the data. Both the authors prepared the manuscript together. G.R.J. supervised the entire project.

## Competing interests

The authors declare no competing interests.
