## [Peer Review File · Nature Communications]

Diamagnetically levitated nanopositioners with large-range and multiple degrees of freedomREVIEWER COMMENTS

Reviewer #1 (Remarks to the Author):

The paper reports a diamagnetically driven, multi-DOF nanopositioner. The design is interesting and novel. However, there are a few issues need clarification.

Technical:

1. To my understanding, stiffness K_x and K_y vary with Z-position. This is a nonlinearity inherent in the system. The proposed positioner has been compared to piezoelectric driven nanopositioner, however, the disadvantage in terms of the nonlinearity vs Z-position are not mentioned. The nonlinearity in stiffness make the positioner less practical in many applications, and this should be clarified in the manuscript.

2. The trade-off between speed/stiffness and travel range is common in nanopositioners. What is the mechanical bandwidth of the proposed nanopositioner in each axis? Frequency responses would be useful to provide a broader understanding of the nanopositioner's dynamics.

3. The drive configurations to achieve multiple DOFs are not reported. How does the nanopositioner, for example, achieve X, Z and θ_z simultaneously? Would one or more DOFs be limited by current limits when moving simultaneously? How does the workspace look like with respect to the maximum currents?

Reviewer #2 (Remarks to the Author):

Excellent paper. The results are noteworthy and push the capabilities of passively levitated machines below 2 nm. The multi-DOF control distinguishes it from much simpler 1 DOF systems. Two very minor comments on the manuscript: (1) the authors might address tilt and any centering issues from the load they encountered (the paragraph above Fig 3 might be an appropriate place), and (2) looks like a typo in the paragraph just below Fig. 3 where they say "it can be drive compliant loads even of large stiffness" - probably "be" should be deleted.

The article doesn't mention (or this reviewer missed it if did; apologies in advance if that was the case) what the system was sitting on. Was it an ordinary lab bench or a vibration isolation platform? If it was an ordinary lab bench then that should be brought out early in article because it is especially significant (in a positive sense) for passively levitated device. If it was a vibration isolation table, that is still OK but it should be brought out clearer.

Manuscript number: NCOMMS-21-41157

Title: Diamagnetically levitated nanopositioners with large-range and multiple degrees of freedom

Authors: K. S. Vikrant and G. R. Jayanth

Paper type: Regular paper

Summary of Responses to the Reviewers

We wish to thank the associate editor and the reviewers for their comments and suggestions. We have addressed all the comments and accordingly made appropriate modifications in the manuscript. The following is the list of important changes made to the manuscript:

(1) In response to the reviewer#1, we have added a paragraph in the main manuscript, and a new section to the supplementary file (Supplementary note 2.6) describing the workspace and the effect of actuation nonlinearities within the workspace.

(2) In response to the reviewer#1, we have added sentences in the main manuscript to discuss about the trade-off between speed/stiffness and travel range in the reported nanopositioner.

(3) In response to the reviewer#1, we have added information about the bandwidth of the multi-zone nanopositioner along all six degrees of freedom in the main manuscript. We have also added a new section to the Supplementary file (Supplementary note 3.2) discussing dynamic modeling, and experimental characterization of the dynamics of nanopositioner along each axis.

(4) In response to the reviewer#1, we have added a paragraph in the main manuscript about the drive configuration to achieve simultaneous multi degree of freedom positioning and a new section to the Supplementary file (Supplementary note 3.1) to derive this in greater detail.

(5) In response to the reviewer#1, we have added a discussion on the dependence of the workspace of the actuator on the current limits in the Supplementary section (Supplementary note 3.1) and commented on the same in the main manuscript.

(6) In response to the reviewer#2, we have added sentences in the main manuscript to discuss about the tilt issues faced from the load.

(7) In response to the reviewer#2, we have corrected the typo in the main manuscript.

(8) In response to the reviewer#2, we have mentioned in the main manuscript that the experimental set-up was mounted on a vibration isolation platform and provided the model number of the platform.

All other comments have also been addressed. The detailed responses are provided on the next page.

Detailed Responses to Reviewer# 1

Comment#1: The paper reports a diamagnetically driven, multi-DOF nanopositioner. The design is interesting and novel. However, there are a few issues need clarification.

Response: We thank the reviewer for the comment. The answer to each of the questions asked by the reviewer is provided below.

Comment#2: To my understanding, stiffness K_x and K_y vary with Z-position. This is a nonlinearity inherent in the system. The proposed positioner has been compared to piezoelectric driven nanopositioner, however, the disadvantage in terms of the nonlinearity vs Z-position are not mentioned. The nonlinearity in stiffness make the positioner less practical in many applications, and this should be clarified in the manuscript.

Response: We thank the reviewer for the comment. The nonlinearity in stiffness is dependent on the extent of displacement of the actuator along the Z-axis from the levitation-plane, and is in turn decided by the height of the workspace above and below the levitation plane. We have added a paragraph in the main manuscript, and a new supplementary note (Supplementary note 2.6) describing the workspace and the change in stiffness along the Z-axis of the workspace.

The workspace along the Z-axis is limited from below by the possibility of contact with the graphite plate, while it is limited from above by the height at which the array becomes unstable. Taken together, the workspace is confined to a height of about $\pm 50 \mu m$ on either side of the levitation plane. Since this is much smaller than the gap between the printed circuit boards ($1502 \mu m$), the effect of actuation nonlinearities would be small within this space. The percentage change in X- and Y-stiffness is calculated to be about 1.4% within the workspace, which is not significant for most practical applications.

However, the Z-stiffness can still change substantially within the workspace, due to the strong dependence of diamagnetic stiffness, and hence the Z-stiffness, on the height above the graphite plate. This can be addressed by sandwiching the array between graphite plates on both the sides, as already discussed in the manuscript, with the air gap being about $51 \mu m$ on both sides. In such a case, the variation in diamagnetic stiffness k_d , and hence that of the Z-stiffness, would be about 0.6% for displacements of about $10 \mu m$ away from the levitation plane but about 38 % over the entire Z- workspace. Thus, in applications where small Z-motion range on the order of $10 \mu m$ is required, this configuration would be suitable. In applications were large motion range is desired along the Z-axis, it is preferable to couple the actuator to a compliant mechanism, such as the trapezoidal flexure reported in the manuscript, and transform the large in-plane motion to corresponding large Z-displacement. In such a situation, the motion of the platform can be restricted to a single plane, viz., the levitation plane, and the variation of stiffnesses with Z-position will not be a matter of concern.

In response to the reviewer, we have added the following sentences in the main manuscript and to the supplementary file to discuss the nonlinearity with respect to Z-position and the way to mitigate its effect:

Main Manuscript, Page No. 5, Paragraph 4, Sentences: 4-10

The workspace below the levitation plane is equal to the air gap, i.e., $z_0 - t_d - t/2$, where t represents the thickness of the magnet array. Above the levitation plane, it is limited by the height z_1 at which $k_z = 0$ (see Supplementary note 2.6), i.e., when $k_d(z_1) = k_x + k_y$. Thus, the workspace along the Z-axis is $z_1 - t_d - t/2$. This is typically much smaller than z_0 since the air gap between the array and the graphite plate is small, and the $k_d(z)$ reduces rapidly for displacements above the graphite plate. Thus, within the workspace, the variation of k_x, k_y along the Z-axis would be small and would not affect most practical applications of the actuator. However, the Z-stiffness can still change substantially within the workspace, due to the strong dependence of $k_d(z)$, and hence k_z , on the height above the graphite plate. Employing another graphite plate beneath the top PCB also helps to substantially reduce the variation in the Z-stiffness within the workspace and would be suitable for applications that require small motion range along the Z-axis.

Main Manuscript, Page No. 7, Paragraph 2, Sentences: 9-10

In applications where large motion range is desired along the Z-axis, it is preferable to employ this actuation strategy since the motion of the platform can be restricted to a single plane, viz., the levitation plane. Thus, the variation of stiffnesses with Z-position will not be a matter of concern.

Main Manuscript, Page No. 9, Paragraph 1, Sentences: 2-3

The workspace below the levitation plane was determined to be $51 \mu\text{m}$ while above the levitation plane it was determined to be $50 \mu\text{m}$ for the case $k_x = k_y = k_d/3$. The percentage change in X- and Y-stiffness was about 1.4% within the workspace, which is small and not significant for most practical applications.

Supplementary File, Page No. 10, Supplementary note 2.6, Paragraph 1, Sentences: 10-12

The height of the workspace along the Z-axis is therefore $z_w = 101 \mu\text{m}$. It is seen that the height of the workspace z_w is much lesser than the gap between the PCBs, viz., $1502 \mu\text{m}$. Thus, the variation in electromagnetic stiffness along the X- and Y-axes is also small within this range, namely, about 1.4%.

Comment#3: The trade-off between speed/stiffness and travel range is common in nanopositioners.

Response: We thank the reviewer for the comment. We agree that there is a trade-off between the speed/stiffness and the travel range for nanopositioners. With the proposed nanopositioner, there is trade-off between the stiffness and travel range along the Z-axis but not along the X-, Y-axes. We have explained this below.

The travel range along the Z-axis above the levitation plane is decided by the height above the levitation plane at which the Z-stiffness becomes zero. Above this height, the array would be unstable. Since the sum of the X-, Y- and Z-stiffnesses equals the diamagnetic stiffness, the Z-stiffness becomes zero when the the sum of X- and Y-stiffnesses equals the diamagnetic stiffness. Since the diamagnetic stiffness reduces for displacements away from the graphite plate, choosing high stiffness k_x, k_y leads to the travel range along the Z-axis to be small.

Along the in-plane axes, the travel range is decided only by the number of meanders in the actuating traces. Thus, the travel range can be increased by employing larger printed circuit boards (PCBs) with larger number of meanders. At the same time, both the in-plane stiffness and response speed are independent of the area of the PCB. The stiffness is dependent on the currents through the traces and the volume of the array. The speed of response along the in-plane axes is determined by the natural frequency, which is decided by the magnitude of currents through the traces and the gap between the PCBs.

Main Manuscript, Page No. 10, Paragraph 1, Sentences: 1-3

It is worth noting that along the X-, Y- axes the motion range is independent of stiffness as well as speed and depends only on the extent of meanders on the PCBs along these axes. Fig. 5(b) shows translation $\Delta z(t)$ within the workspace along the Z-axis achieved by controlling $I_y^{u1} - I_y^{l1}$. Along the Z-axis, there exists a trade-off between the size of the workspace and stiffness, wherein higher X- and Y-stiffnesses leads to reduction in workspace along the Z-axis.

Comment#4: What is the mechanical bandwidth of the proposed nanopositioner in each axis? Frequency responses would be useful to provide a broader understanding of the nanopositioner's dynamics.

Response: We thank the reviewer for the comment. We have added information about the mechanical bandwidth of the multi-zone nanopositioner along all six degrees of freedom in the main manuscript. We have also added a new supplementary note (Supplementary note 3.2), discussing dynamic modeling, and experimental characterization of the dynamics of nanopositioner along each axis. We have used the experimental results to report the frequency responses in the supplementary note.

The text added to the main manuscript and the new supplementary note are both pasted below.

Main Manuscript, Page No. 10, Paragraph 1, Sentences: 11-14

From the view-point of dynamics, the magnet array behaves as a rigid body that is electromagnetically trapped with the trap stiffness being k_x, k_y and k_z along X-, Y- and Z- axes respectively. Thus, it can be modeled as a mass-spring-damper system, where the damping arises due to the surrounding air. This model was validated experimentally and the bandwidth of the nanopositioner along the linear channels was evaluated from the step responses along each axis. The open-loop bandwidth along X-, Y- and Z-axes were $\omega_{bx} = \omega_{by} = 131 \text{ rad/s}$, $\omega_{bz} = 153 \text{ rad/s}$, while the bandwidth for rotations about X-, Y- and Z-axes were $\omega_{b\theta_x} = \omega_{b\theta_y} = 121 \text{ rad/s}$ and $\omega_{b\theta_z} = 90 \text{ rad/s}$ (see Supplementary note 3.2).

Supplementary File, Page No. 13-16, Supplementary note 3.2

3.2 Dynamic modeling and characterization of the nanopositioner

In equilibrium, each magnet in the array is electromagnetically trapped with trap stiffness being k_x, k_y and k_z along X-, Y- and Z-axes respectively. Thus, for small displacements away from equilibrium, the dynamic behavior would be that of a mass-spring-damper

system, where the damping arises due to the surrounding air. If m_a represents the mass of the overall array, and b_x represents the damping coefficient, the dynamic model along the X-axis would be

$$(74) \quad m_a \ddot{x} + b_x \dot{x} + k_x x = f_x,$$

where, f_x represents the force along the X-axis. Similar equations would be valid for motion along Y- and Z-axes as well.

A linear second-order system model would also be valid for angular dynamics about X-, Y- and Z-axis as well, provided the angular displacements are small. The model along the X-axis is given to be

$$(75) \quad I_{xx} \ddot{\theta}_x + b_\theta \dot{\theta}_x + k_{\theta x} \theta_x = \tau_x,$$

where, I_{xx} represents the moment of inertia of the array about the X-axis, b_θ represents the angular damping coefficient and $k_{\theta x}$ represents the angular stiffness about the X-axis, while τ_x represents the torque. Similar equations would be valid for small rotations about the Y- and Z-axes as well.

The angular stiffness $k_{\theta x}$ can be obtained by noting that if x_i represents the offset of the center of the i^{th} magnet in the array, a small angular displacement θ_x results in a Z-displacement of $\theta_x x_i$ and causes the array to experience a restoring force of $-k_z \theta_x x_i$. The moment of this force about the X-axis is given by $-k_z \theta_x x_i^2$. Thus, the torsional stiffness is given to be

$$(76) \quad k_{\theta x} = k_z \sum_i x_i^2.$$

Likewise, the angular stiffness about Y-axis is given by,

$$k_{\theta y} = k_z \sum_i y_i^2,$$

while the Z-angular stiffness is obtained to be

$$(77) \quad k_{\theta z} = k_x \sum_i y_i^2 + k_y \sum_i x_i^2.$$

These analyses reveal that the dynamic model of the actuator for small displacements is that of a second order system along all its degrees of freedom. This was validated experimentally from the step responses obtained along the linear and angular channels and the bandwidth of the nanopositioner along the X-, and Z- linear channels and θ_x , θ_z angular channels was evaluated from these responses. In the experiments, the X- and Y-stiffnesses were chosen to be the same. Thus, by symmetry, the linear dynamic responses along X- and Y-axes would be identical as also the angular dynamic responses about X- and Y-axes.

Supplementary Fig. 6 (a)-(d) plots the step responses along the four axes and also the fitted second order models, whose coefficients were chosen to match the experimental step responses. The insets in each figure show the responses in the vicinity of time $t = 0$. It is seen from the figures that the step responses of the second order models match the experimental responses well. The natural frequencies and the damping factors for translation along X-, Y- and Z-axes are $\omega_{nx} = \omega_{ny} = 130.8 \text{ rad/s}$, $\omega_{nz} = 140.7 \text{ rad/s}$, $\xi_x = \xi_y = 0.005$ and $\xi_z = 0.10$. Similarly the natural frequencies and damping factors

for rotations about X-, Y- and Z-axes are $\omega_{n\theta_x} = \omega_{n\theta_y} = 110.8 \text{ rad/s}$, $\omega_{n\theta_z} = 89.7 \text{ rad/s}$, $\xi_{\theta_x} = \xi_{\theta_y} = 0.12$ and $\xi_{\theta_z} = 0.007$. The damping factor is seen smaller, and hence the settling time is larger, for actuation resulting in out-of-plane displacements compared to actuation resulting in in-plane displacements. This may be attributed to the squeeze-film damping effect arising due to the narrow air gap between the graphite plate and the array.

Supplementary Fig. 7 (a)-(d) plot the magnitude responses of the fitted second-order models along each axis and shows that the open-loop bandwidth is $\omega_{bx} = \omega_{by} = 131 \text{ rad/s}$, $\omega_{bz} = 153 \text{ rad/s}$, $\omega_{b\theta_z} = 90 \text{ rad/s}$ and $\omega_{b\theta_x} = \omega_{b\theta_y} = 121 \text{ rad/s}$.

Supplementary Fig. 6. Plots showing the step response along the four axes: (a) X- axis, (b) Z-axis, (c) θ_z - axis and (d) θ_x - axis. The measurements have been normalized with respect to their steady-state values x_{ss} , z_{ss} , θ_{xss} and θ_{zss} respectively. Initially the currents through all the traces of the multi-zone positioner were maintained constant at 200 mA. Next the currents through appropriate traces were increased by 10 mA to achieve the required translational and rotational step responses.

Supplementary Fig. 7. Plots showing the magnitude response of the fitted second order model along the four axes: (a) X- axis, (b) Z-axis, (c) θ_x - axis and (d) θ_z - axis.

Comment#5: The drive configurations to achieve multiple DOFs are not reported. How does the nanopositioner, for example, achieve X, Z and thetaZ simultaneously?

Response: We thank the reviewer for the comments. We have now added a paragraph in the main manuscript about the drive configuration to achieve simultaneous multi degree of freedom positioning and a new supplementary note (Supplementary note 3.1) to derive this in greater detail. The important steps are briefly described here. The detailed procedure reported in the supplementary note is reproduced, along with the changes made to the manuscript.

To obtain the drive configuration, i.e., the currents necessary for simultaneous multi-degree-of-freedom positioning, it is first noted that within the pitch p of a single meander, there is a unique relationship between the loads and the actuation currents and that this repeats itself with a periodicity of p along the X- and Y-axes. This fact is employed to obtain the drive configurations to achieve the desired multi-degree of freedom position in three steps: In the first step, the specified X- and Y-displacements and rotation about the Z-axis are achieved. In the second step, the loads that need to be applied to achieve the specified positions along the remaining degrees of freedom are obtained. In the third step, the desired X- and Y-stiffnesses are also specified and the actuation currents necessary to apply the loads and achieve the specified stiffnesses at the specified X-,Y-position are obtained using Eqns. (2-3) given in the main manuscript. The resulting relationship between the necessary currents and the specified displacements are described in Supplementary note 3.1.

This procedure can also be employed to move along X-, Z- and θ_z simultaneously. In this case, the desired X- and Y-stiffnesses are chosen, the array is trapped at a fixed Y-position. Likewise,

the torque necessary for rotation about X- and Y-axes are both set to zero. The X- position is changed by the desired amount, followed by rotation by θ_z and finally the array is displaced by the desired extent along the Z-axis by electromagnetically applying the necessary Z-force to the actuator.

Main Manuscript, Page No. 7, Paragraph 2, Sentences: 1-7

A noteworthy capability of the multi-zone nano-positioning system is that it also enables simultaneous positioning along all the six degrees of freedom. To obtain the drive configuration, i.e., the currents necessary for simultaneous multi-degree-of-freedom positioning, it is first noted that within the pitch p of a single meander, there is a unique relationship between the loads and the actuation currents and that this repeats itself with a periodicity of p along the X- and Y-axes. This fact is employed to obtain the drive configurations to achieve the desired multi-degree of freedom position in three steps: In the first step, the specified X- and Y-displacements and rotation about the Z-axis are achieved. In the second step, the loads that need to be applied to achieve the specified Z-position, and rotation about X- and Y-axes are obtained. In the third step, the X- and Y-stiffnesses are also specified and the actuation currents necessary to apply the loads and achieve the specified stiffnesses at the specified X-, Y-position are obtained using Eqns. (2), (3). The resulting relationship between the necessary currents and the specified displacements are described in Supplementary note 3.1.

Supplementary File, Page No. 11-13, Supplementary note 3.1

3.1 Drive configuration to achieve multi-degree of freedom positioning

This section describes the procedure and the currents necessary to achieve simultaneous positioning of the nanopositioner along all the six degrees of freedom. It is noted that within the pitch p of a single meander, there is a unique relationship between the loads and the actuation currents and that this repeats itself with a periodicity of p along the X- and Y- axes. This fact is employed to obtain the drive configurations to achieve the desired multi-degree of freedom position in three steps: In the first step, the specified linear displacement along X- and Y-axes x_s, y_s and angular displacement about the Z-axis θ_{zs} are achieved. In the second step, the loads that need to be applied to achieve the specified Z- displacement z_s and angular displacements about X- and Y-axes θ_{xs}, θ_{ys} are obtained. In the third step, the X- and Y-stiffnesses k_{xs}, k_{ys} are also specified and the actuation currents necessary to apply the loads and achieve the stiffnesses at the specified position are obtained using Eqns. (43-44) derived in the Supplementary note 2.3. These steps are elaborated below.

In the first step, the desired linear displacement along X- and Y-axes x_s, y_s and angular displacement about the Z-axis θ_{zs} are achieved by suitably shifting the equilibrium points of the magnet array in each of the zones of the actuator. To achieve linear displacements x_s, y_s the equilibrium points in all of the zones are shifted along the X- and Y-axes with speeds v_x, v_y respectively, and for durations $t_x = x_s/v_x$ and $t_y = y_s/v_y$ respectively. This is achieved by applying the current waveforms:

$$I_y^{l1} = I_y^{u1} = I_{0x} \sin\left(\frac{2\pi}{p} v_x t\right) \quad ,$$

(60)

$$I_y^{l2} = I_y^{u2} = I_{0x} \cos\left(\frac{2\pi}{p} v_x t\right) \quad , \quad (61)$$

$$I_x^{l1} = I_x^{u1} = I_{0y} \sin\left(\frac{2\pi}{p} v_y t\right) \quad , \quad (62)$$

$$I_x^{l2} = I_x^{u2} = I_{0y} \cos\left(\frac{2\pi}{p} v_y t\right) \quad . \quad (63)$$

In Eqns. (60)-(63), I_{0x} , I_{0y} are chosen to be $I_{0x} = \frac{p^2}{8\sqrt{2}\pi^2 m' b_{1zy}} k_x$ and $I_{0y} = \frac{p^2}{8\sqrt{2}\pi^2 m' b_{1zy}} k_y$. This ensures that the specified trap stiffness is maintained during the displacement.

To achieve angular displacement θ_{zs} , the equilibrium points in diagonally opposite zones were shifted by equal and opposite amounts tangential to the array. For small θ_{zs} , this is given by $l_a \theta_{zs}$ where l_a represents the distance of the center of the arm from the axis of rotation. The X- and Y- waveforms are similar to those described in Eqns. (60)-(63). The sign of X- and Y-displacements in each zone is chosen to ensure that the array moves tangential to the center of rotation and ensures rotation of the array about the Z-axis by the desired amount. For large rotations, two issues are introduced from the view-point of analysis: first, the locations of the centers of the magnets will no longer be at the desired pitch along X- and Y-axes, and second, the in-plane rotation of the magnets changes the resulting point dipole m' . While the drive configuration for large angle rotation can also be derived for this case, a more appropriate arrangement of traces to rotate by large angles would be to employ traces patterned in the radial and azimuthal directions on the PCB.

In the second step, the specified position of the center of the actuator along the Z-axis is given to be z_s and desired orientations about X- and Y-axes are given to be θ_{xs} and θ_{ys} respectively. If x_i and y_i represent the X- and Y-coordinates of the center of the i^{th} magnet in the array, the height of the magnet above the levitation plane is given by

$$z_i = z_s - \theta_{xs} y_i + \theta_{ys} x_i. \quad (64)$$

The Z-force necessary to position the array in this configuration is given by $[W - \sum_i F_d(z_i)]$. Thus, the load on each arm would be $1/4^{th}$ of this value. Further in each arm, this force is applied jointly by the X- and Y- traces. Thus, both X- and Y- traces has to apply force F_{zs} given by:

$$F_{zs} = 0.125[W - \sum_i F_d(z_i)]. \quad (65)$$

The necessary torque about X- and Y-axes on each arm is given by $\boldsymbol{\tau}_s = \sum_i \mathbf{r}_i \times (\mathbf{F}_d(z_i))$, where $\mathbf{r}_i = [x_i \ y_i \ 0]^T$ and $\mathbf{F}_d(z_i) = [0 \ 0 \ F_d(z_i)]^T$. The resulting X-, Y-components of $\boldsymbol{\tau}_s$ are given by

$$\tau_{xs} = 0.25 \sum_i F_d(z_i) y_i, \quad (66)$$

$$\tau_{ys} = -0.25 \sum_i F_d(z_i) x_i. \quad (67)$$

Eqns. (64)-(67) specify all the loads necessary for achieving the desired orientation and Z-position.

In the third step, the necessary currents in each of the zones are obtained by using Eqn. (1) in the paper. If x_j , y_j represent the X- and Y-displacements of the equilibrium in zone j of the actuator ($j = 1,2,3,4$), the resulting currents for this zone are given by

$$\begin{bmatrix} I_y^{l1} \\ I_y^{l2} \\ I_y^{u1} \\ I_y^{u2} \end{bmatrix} = \begin{bmatrix} A_y^{l1}(x_j) & B_y^{l1}(x_j) & C_y^{l1}(x_j) \\ A_y^{l2}(x_j) & B_y^{l2}(x_j) & C_y^{l2}(x_j) \\ A_y^{u1}(x_j) & B_y^{u1}(x_j) & C_y^{u1}(x_j) \\ A_y^{u2}(x_j) & B_y^{u2}(x_j) & C_y^{u2}(x_j) \end{bmatrix} \begin{bmatrix} \tau_{ys} \\ F_{zs} \\ k_{xs} \end{bmatrix} \quad (68)$$

$$\begin{bmatrix} I_x^{l1} \\ I_x^{l2} \\ I_x^{u1} \\ I_x^{u2} \end{bmatrix} = \begin{bmatrix} A_x^{l1}(y_j) & B_x^{l1}(y_j) & C_x^{l1}(y_j) \\ A_x^{l2}(y_j) & B_x^{l2}(y_j) & C_x^{l2}(y_j) \\ A_x^{u1}(y_j) & B_x^{u1}(y_j) & C_x^{u1}(y_j) \\ A_x^{u2}(y_j) & B_x^{u2}(y_j) & C_x^{u2}(y_j) \end{bmatrix} \begin{bmatrix} \tau_{xs} \\ F_{zs} \\ k_{ys} \end{bmatrix} \quad (69)$$

Where, the coefficients of the matrices are given to be

$$\begin{aligned} A_y^{l1}(x_j) &= -A_y^{u1}(x_j) = \frac{1}{2m'b_{1x}} \cos\left(\frac{2\pi x_j}{p}\right), & A_x^{l1}(y_j) &= -A_x^{u1}(y_j) = \frac{1}{2m'b_{1x}} \cos\left(\frac{2\pi y_j}{p}\right), \\ A_y^{l2}(x_j) &= -A_y^{u2}(x_j) = \frac{1}{2m'b_{1x}} \sin\left(\frac{2\pi x_j}{p}\right), & A_x^{l2}(y_j) &= -A_x^{u2}(y_j) = \frac{1}{2m'b_{1x}} \sin\left(\frac{2\pi y_j}{p}\right), \\ B_y^{l1}(x_j) &= -B_y^{u1}(x_j) = \frac{p}{4\pi m'b_{1x}} \sin\left(\frac{2\pi x_j}{p}\right), & B_x^{l1}(y_j) &= -B_x^{u1}(y_j) = \frac{p}{4\pi m'b_{1x}} \sin\left(\frac{2\pi y_j}{p}\right) \\ B_y^{l2}(x_j) &= -B_y^{u2}(x_j) = -\frac{p}{4\pi m'b_{1x}} \cos\left(\frac{2\pi x_j}{p}\right), & B_y^{l2}(y_j) &= -B_y^{u2}(y_j) = \\ & -\frac{p}{4\pi m'b_{1x}} \cos\left(\frac{2\pi y_j}{p}\right), & C_y^{l1}(x_j) &= C_y^{u1}(x_j) = -\frac{p^2}{8\pi^2 m'b_{1zy}} \sin\left(\frac{2\pi x_j}{p}\right), & C_y^{l1}(y_j) &= \\ C_y^{u1}(y_j) &= -\frac{p^2}{8\pi^2 m'b_{1zy}} \sin\left(\frac{2\pi y_j}{p}\right), & C_y^{l2}(x_j) &= C_y^{u2}(x_j) = \frac{p^2}{8\pi^2 m'b_{1zy}} \cos\left(\frac{2\pi x_j}{p}\right), \\ & & C_y^{l2}(y_j) &= C_y^{u2}(y_j) = \frac{p^2}{8\pi^2 m'b_{1zy}} \cos\left(\frac{2\pi y_j}{p}\right). \end{aligned}$$

It is noted that , $x_j = x_s - \frac{l_a \theta_{zs}}{\sqrt{2}}$ for zone numbers $j = 1,2$ and $x_j = x_s + \frac{l_a \theta_{zs}}{\sqrt{2}}$ for zone numbers $j = 3,4$. Likewise $y_j = y_s + \frac{l_a \theta_{zs}}{\sqrt{2}}$ for zone numbers $j = 1,4$ and $y_j = y_s - \frac{l_a \theta_{zs}}{\sqrt{2}}$ for zone numbers $j = 2,3$.

Eqn. (68) and (69) specify the drive configuration of the nanopositioner to simultaneously displace the positioner by the specified amount along all the six degrees of freedom.

Comment#6: Would one or more DOFs be limited by current limits when moving simultaneously? How does the workspace look like with respect to the maximum currents?

Response: We thank the reviewer for the comment. We have added a discussion on the workspace of the actuator in Supplementary note 2.6 and commented on the same in the main manuscript. A limit on the maximum current does potentially limit the workspace, and we have discussed the procedure to obtain this workspace in supplementary note 3.1. However, for the

actual maximum current that can be applied, the associated workspace is much larger than that determined by the following two factors the spatial constraint due to the graphite plate and instability along the Z-axis beyond a certain distance above the plate. Thus, the actual workspace is determined only by these two factors, which, when taken together, limit the workspace to a total height of $101\mu\text{m}$ along the Z-axis. The area of the workspace along X- and Y- axes is determined by the area of the PCBs covered by the meanders along each axis. The currents necessary to simultaneously position the actuator in any desired multi-degree-of-freedom configuration within this workspace is obtained from the drive configurations described in Supplementary note 3.1. Using these equations, the maximum value of the currents necessary for multi-degree-of-freedom positioning is found to be about 245 mA, i.e., about 8 times lesser than the actual maximum current $I_{max} = 2A$. This verifies that specified I_{max} , workspace is determined by the factors described in Supplementary note 2.6 and not by the maximum current.

Main Manuscript, Page No. 9, Paragraph 1, Sentences: 4-6

The currents necessary to position the actuator anywhere in this workspace can be obtained from the equations for the drive configuration. Numerically, the currents were found to be about 245 mA, which is much lesser than the maximum current I_{max} . This verifies that for the chosen parameters, the current limits do not influence the workspace of the device.

Supplementary File, Page No.13, Supplementary note 3.1

Eqn. (68) and (69) specify the drive configuration of the nanopositioner to simultaneously displace the positioner by the specified amount along all the six degrees of freedom. These equations can be employed to compute the maximum value of currents for the specified loads and stiffnesses. Since I_y^{l1} is given by $\frac{\tau_{ys}}{2m'b_{1x}} \cos\left(\frac{2\pi x}{p}\right) + \left(\frac{pF_{zs}}{4\pi m'b_{1x}} - \frac{p^2 k_{xs}}{8\pi^2 m'b_{1zy}}\right) \sin\left(\frac{2\pi x}{p}\right)$, its maximum value is $\sqrt{\left(\frac{\tau_{ys}}{2m'b_{1x}}\right)^2 + \left(\frac{pF_{zs}}{4\pi m'b_{1x}} - \frac{p^2 k_{xs}}{8\pi^2 m'b_{1zy}}\right)^2}$. Following this procedure, the maximum value of all the currents is given by

$$I_{ymax}^{l1} = I_{ymax}^{l2} = \sqrt{\left(\frac{\tau_{ys}}{2m'b_{1x}}\right)^2 + \left(\frac{pF_{zs}}{4\pi m'b_{1x}} - \frac{p^2 k_{xs}}{8\pi^2 m'b_{1zy}}\right)^2}, \quad (70)$$

$$I_{ymax}^{u1} = I_{ymax}^{u2} = \sqrt{\left(\frac{\tau_{ys}}{2m'b_{1x}}\right)^2 + \left(\frac{pF_{zs}}{4\pi m'b_{1x}} + \frac{p^2 k_{xs}}{8\pi^2 m'b_{1zy}}\right)^2}, \quad (71)$$

$$I_{xmax}^{l1} = I_{xmax}^{l2} = \sqrt{\left(\frac{\tau_{xs}}{2m'b_{1x}}\right)^2 + \left(\frac{pF_{zs}}{4\pi m'b_{1x}} - \frac{p^2 k_{ys}}{8\pi^2 m'b_{1zy}}\right)^2}, \quad (72)$$

$$I_{xmax}^{u1} = I_{xmax}^{u2} = \sqrt{\left(\frac{\tau_{xs}}{2m'b_{1x}}\right)^2 + \left(\frac{pF_{zs}}{4\pi m'b_{1x}} + \frac{p^2 k_{ys}}{8\pi^2 m'b_{1zy}}\right)^2}. \quad (73)$$

Since the loads in Eqns. (70)-(73) are functions of the specified displacements along the different degrees of freedom, these equations can be employed to obtain the workspace as determined by the current limit I_{max} alone. However, upon computing the currents within the workspace determined by spatial constraints and instability, as discussed in Supplementary note 2.6, the maximum value of current is found to be 245 mA, which is much less than $I_{max} = 2A$. Thus, this indicates that for the specified I_{max} , workspace is determined by the factors described in Supplementary note 2.6 and not by the current limit.

Detailed Responses to Reviewer# 2

Comment#1: Excellent paper. The results are noteworthy and push the capabilities of passively levitated machines below 2 nm. The multi-DOF control distinguishes it from much simpler 1 DOF systems.

Response: We appreciate the reviewer's comments and are happy that the reviewer finds our work interesting. The answers to the reviewers questions are provided below.

Comment#2: Two very minor comments on the manuscript: (1) the authors might address tilt and any centering issues from the load they encountered (the paragraph above Fig 3 might be an appropriate place), and (2) looks like a typo in the paragraph just below Fig. 3 where they say "it can be drive compliant loads even of large stiffness" - probably "be" should be deleted.

Response: We thank the reviewer for the comment. We have added a few sentences in the main manuscript to clarify the tilt and centering issues faced from the load. Further we have corrected the typo. The added and modified sentences are shown below:

Main Manuscript, Page Number: 8, Paragraph 1, Sentence 17

The payload was placed in the center of the stage to avoid any tilt due to the uneven distribution of the weight of the payload.

Main Manuscript, Page Number: 6, Paragraph 2, Sentence 4

Third, in combination with position feedback, it can drive compliant loads even of large stiffness.

Comment#3: The article doesn't mention (or this reviewer missed it if did; apologies in advance if that was the case) what the system was sitting on. Was it an ordinary lab bench or a vibration isolation platform? If it was an ordinary lab bench then that should be brought out early in article

because it is especially significant (in a positive sense) for passively levitated device. If it was a vibration isolation table, that is still OK but it should be brought out clearer.

Response: We thank the reviewer for the comment. We have a sentence in the main manuscript to clarify that the positioner was mounted on a Vibration isolation table. The added sentence is shown below:

Main Manuscript, Page Number: 7, Paragraph 3, Sentence 10

The entire set-up including the positioner and the microscopes were mounted on a vibration isolation platform (HOLMARC, PVISA 180-120).

REVIEWERS' COMMENTS

Reviewer #1 (Remarks to the Author):

All comments have been addressed in a satisfactory manner. I have no further questions.

Reviewer #2 (Remarks to the Author):

This is a follow-up review based on author changes from reviewer previous comments. The authors have addressed this reviewer's comments satisfactorily.

Manuscript number: NCOMMS-21-41157A

Title: Diamagnetically levitated nanopositioners with large-range and multiple degrees of freedom

Authors: K. S. Vikrant and G. R. Jayanth

Paper type: Regular paper

Summary of Response to the Reviewers

We wish to thank the associate editor and the reviewers for their comments. The responses to the reviewers are provided below.

Response to Reviewer# 1

Comment: All comments have been addressed in a satisfactory manner. I have no further questions.

Response: We are pleased that the reviewer found our responses satisfactory, and we would like to thank the reviewer for the time and effort.

Response to Reviewer# 2

Comment: This is a follow-up review based on author changes from reviewer previous comments. The authors have addressed this reviewer's comments satisfactorily.

Response: We are pleased that the reviewer found our responses satisfactory, and we would like to thank the reviewer for the time and effort.